# Compact and Polarization Insensitive Satellite Band Perfect Metamaterial Absorber for Effective Electromagnetic Communication System

**DOI:** 10.3390/ma16134776

**Published:** 2023-07-02

**Authors:** Md. Salah Uddin Afsar, Mohammad Rashed Iqbal Faruque, Sabirin Abdullah, K. S. Al-Mugren

**Affiliations:** 1Space Science Centre (ANGKASA), Institute of Climate Change (IPI), Universiti Kebangsaan Malaysia, Bangi 43600, Malaysia; p108627@siswa.ukm.edu.my (M.S.U.A.); dr_sabirin@ukm.edu.my (S.A.); 2Directorate of Secondary and Higher Education Bangladesh (DSHE), Dhaka 1000, Bangladesh; 3Physics Department, Science College, Princess Nourah bint Abdulrahman University, Riyadh 11671, Saudi Arabia; ksalmogren@pnu.edu.sa

**Keywords:** electrical conductivity, permeability, satellite band, wave guide

## Abstract

A commercially viable metal–dielectric–metal configured triple-band metamaterial absorber is offered in this paper. It is an aggregation of four compact symmetric circles, with a swastika-shaped metal structure, which are bonded by two split-ring resonators (SRRs). Copper (annealed) of electrical conductivity 5.8 × 10^7^ Sm^−1^ is used for the ground plate and resonator portion of the top layer and an FR 4 dielectric of permittivity 4.3 is used as a substrate. The structural parameters of the unit cell were determined by a trial and error method. FIT-based 3D simulation software (CST microwave studio, 2019 version was used to characterize the proposed perfect metamaterial absorber (PMA). Three resonance peaks were observed at frequencies 3.03, 5.83 and 7.23 GHz with an absorbance of 99.84%, 99.03% and 98.26%, respectively. The numerical result has been validated by some authentic validation methods. Finally, a microwave network analyzer (PNA) of Agilent N5227 with waveguide ports were deployed for measurement. The simulation and experimental results show better harmony. The proposed PMA has a unique design and a small dimension with higher absorption compared to other contemporary studies. This special type of polarization, insensitive S- and C-band PMA, is designed for a telecommunication system via full-time raw satellite and radar feeds.

## 1. Introduction

Every material has some natural properties in built in it. An artificial structure that generates some unnatural or peculiar characteristics is commonly termed as metamaterial [1,2,3]. In recent years, the research in this field has been growing rapidly due to its potential uses. A metamaterial absorber (MA) is a device that absorbs incident electromagnetic radiation at it’s operating frequency [4]. Pendry’s theory gave the impression of the first left handed metamaterial (LHM) of Smith’s, which consisted of metallic SRR and thin wire [5,6]. An important use of metamaterial is the metamaterial absorber, which has become of great interest among scientists in the last decade. Due to its novel and excellent characteristics, it has been deployed in the microwave region [7], terahertz region [8], infrared [9] and visible wavelength region [10]. Liang et al. offered an anisotropic plasmonic metasurface which consisted of arrays of vertically oriented double-pillar metamolecules covered by a thin layer of gold [11]. It showed a strong response in the mid-infrared spectrum, which could be applied for the enhancement of light–matter interaction for nano-optics and quantum photonics. Moreover, a mid-infrared detector-based dual-band MMA was suggested by Chen et al. in 2021 for aerospace and military field applications [12]. This theoretical MMA showed perfect absorption in the numerical analysis.

Meanwhile, a wavelength tunable absorption control (WTAC) in thermal bolometers was offered in [13]. It achieved high responsivity through paired bandwidth tailoring. A triple-band metamaterial absorber was designed by Wu et al. for electromagnetic shielding and energy harvesting [14]. In the same year, a thin circular sector metallic resonator-based metamaterial absorber was published in [15]. This MA showed above 98% absorption in its minor resonance frequency of 3.35 GHz. A curtailed cone-shaped dual-band MA comprising an array of periodic meta-atoms was reported in [16] by Kim et al. CST-simulated results exhibited a 90% EM absorption in the microwave region. A single-peaked polarization-sensitive absorber, consisting of a single square resonator on FR 4 dielectric substrate, was published in 2016 [17]. The benefits of the above discussed MA compared to the traditional absorbers are cost effectiveness, ease of configuration and light weight. To ensure the compactness and perfect absorption from any electromagnetic MA, the material permittivity (ℇ) and permeability (µ) were adjusted to match the free impedance with the input impedance [18]. Moreover, dual-band and multiband, with independent polarization and a wide-angle incidence absorber, is more advantageous than a single-band low-absorption MA. Therefore, the researchers have given more attention to developing an efficient multiband metamaterial absorber for multi-purpose use. A dual-band broadband absorber was designed in [19] that showed two resonance peaks at 8.5 GHz and 15.50 GHz with 70% and 90% absorptivity, respectively. A lump resistor-loaded periodic arrangement of a circular ring-based metamaterial absorber was presented by Nguyen et al. [20]. It exhibited dual resonance peaks at 7.8 and 12.6 GHz with absorption at more than 90%. To show the EMI shielding performance, Zhang et al. offered an MA which absorbed 90% of incident EM radiation in the microwave range [21]. An ultra-thin polarization-insensitive triple-band MA was designed with a Jerusalem cross and concentric circular metallic resonators in [22]. In numerical simulations, it showed three random resonance peaks at 4.4, 6.05 and 13.9 GHz with over 90% absorption. The 3D and flatland toroidal metastructures are designated in [23]. These revolutionary tools have been executed for infrared photodetectors, immune-biosensors, ultraviolet beam sources and waveguides. To claim a metamaterial as an absorber, the reflection and transmission would be zero, as a+r+t=1. This can be achieved by increasing the absorption of the proposed structure [24]. A metallic ring-structured resonator-based MMA was suggested by Qu et al. in 2021 for EM wave absorption in a microwave regime [25]. It exhibited nearly 99% absorption in the 4–30 GHz frequency range. In [26], three modified hexagonal cell-based T-shaped triple-band MA were presented. A dielectric FR 4 of 1mm thickness was used as a substrate, and it exhibited three resonance peaks at 5.8, 9.52 and 11.97 GHz with an absorbance of 95%, 97% and 99%, respectively.

In this study, a novel triple-band polarization independent perfect metamaterial absorber is presented. The electrical dimensions of the unit cell are 0.101λ × 0.101λ, which is very compact in size. The absorption peaks are influenced by the geometric parameters of the proposed design, which are observed at the electrical wavelengths λ_1_ = 99.009 mm, λ_2_ = 51.457 mm and λ_3_ = 41.493 mm with an absorbance of 99.84%, 99.03% and 98.26%, respectively. The proposed PMA has the benefits of excellent triple-band unity absorption efficiency, better operating angle polarization forbearance and expedient manufacturing that allow it to be a good candidate for a satellite-dependent telecommunication system.

## 2. Design and Methodology

### 2.1. Structural Details of the Unit Cell

The dimension (L × B × t) of the PMA unit cell was 10 × 10 × 1.6 mm^3^. A copper plate of 0.035 mm thickness was used to constrain the transmission of EM radiation. The parameters of the proposed PMA with symbols are depicted in Figure 1a, whereas Figure 1b shows the oblique view of the unit cell. A flame retardant dielectric FR 4 of 1.6 mm thickness was sandwiched as a substrate between the bottom layer and top resonator level.

The top layer of the unit cell consisted of four symmetrical swastika-shaped resonators which were surrounded by a circular split-ring resonator (CSRR) along with two split-ring resonators. The width (a) and split gap (G) of each SRR was 0.50 mm. The inner radius (r_1_) and outer radius (r_2_) of the CSRR was 1.5 mm and 1.7 mm, and the split gap (g) was 0.30 mm. The length of the handle and head of the swastika shapes were L_1_ = 2 mm and L_2_ = 0.80 mm, respectively, which is demonstrated in Figure 1c. The thickness (t) of the bottom metal sheet and all resonators was 0.035 mm. Figure 1d represents the simulation view of the PMA unit cell. The brief geometry of the proposed metamaterial absorber is shown in Table 1. 

### 2.2. Design Methodology

To investigate the characteristics and performance of the proposed metamaterial absorber, it was simulated by the three-dimensional finite integration technique-based EM simulation solver CST microwave studio [27]. Proper boundary condition is essential to extract the desired outcome from the unit cell. The unit cell boundary condition was applied along the x-axis and y-axis and incident EM radiation was propagated parallel to the z-axis in this regard [28]. Two wave guide ports were placed at the two ends of the z-axis. Three concerned important incident-reflection, and transmission and absorption occurred when EM radiation passed through the z-axis. The surface input electromagnetic radiation was agitated with the interface of the material. The wave particle photons interacted with each other, as well as with the medium, and affected the transmission exponentially. Resonance happened when the frequency of the incident wave became closer to the surface frequency. It reduced the reflection of the incident EM wave and therefore the radiation was absorbed by the medium [29]. The absorption efficiency of a metamaterial absorber basically depends on the reflection coefficient, *R*(*ω*), and transmission coefficient, *T*(*ω*), which can be related by Equation (1) [30,31].
(1)$$A\left(\omega \right)+R(\omega )+T(\omega )=1$$

For the minimum value of *R*(*ω*) and *T*(*ω*), the absorption value will be maximum, i.e.,

if *R*(*ω*) = *R*(*ω*)*_min_* and *T*(*ω*) = *T*(*ω*)*_min_* then *A*(*ω*) = *A*(*ω*)*_max_*.

A copper plate of 0.035 mm thickness was used as a bottom plate to restrain the transmission of EM wave. Therefore, the absorption efficiency is the only function of the reflection coefficient. Finally, absorption efficiency is as follows:(2)A(ω)max=1−R(ω)min

## 3. Results and Discussion

Figure 2 depicts the numerical results of the reflection coefficient (S_11_), transmission coefficient (S_21_) and absorption rate. It can be observed from the graph that three absorption peaks were seen at 3.03, 5.83 and 7.23 GHz with magnitude −27.97 dB,−20.13 dB and −17.56 dB. The proposed PMA covers S- and C-band with an absorption of 99.84%, 99.03% and 98.26%, along with standard bandwidths. The transmission coefficient (S_21_) is zero, which is indicated by the red in the graph. The polarization and incident angle independency was an important issue to make an absorber confine free. In this section, polarization and incident angle independency will be investigated.

The top layer’s patches of the unit cell were rotated from normal position (0°) to 90°. The numerical absorption results for both positions show very close resemblance to each other, which was the cause of the polarization insensitiveness. The absorption results at 0° and 90° are depicted in Figure 3.

The CST results for different oblique incident angles are illustrated in Figure 4. For normal incidence (θ = 0°), three absorption peaks were observed at 3.03, 5.83 and 7.23 GHz, with nearly unity absorption. At θ = 15°, nearly the same absorption was shown at three resonance frequencies: 3.0, 5.69 and 6.90 GHz. Again, for the incident angle θ = 30°, it also shows nearly unity absorbance at the three resonance peaks: 3.0, 5.78 and 7.02 GHz. Similarly, at the incident angle θ = 45°, three resonance peaks were noticed at 3.02, 5.84 and 7.03 GHz, with an absorbance of 95.62%, 99.03% and 97.40%, respectively. Moreover, three resonance peaks were observed at 7.03, 5.90 and 7.05 GHz, with 84.35%, 91.02% and 93.30% for the 60° angle. This tendency continued up to 75° with some variation in absorbance. Thus, the suggested PMA is a wide-angle incidence up to 75°. We have observed that there was a small amount of variation in absorption results for different incident angles. Absorption performance was investigated for every 15° of incident angle. The proposed design structure was assembled by the addition and subtraction of different shapes. This is due to the fact that configurable changes after 15° of variation of the incident angle can affect the absorption results.

The performance of the PMA was also investigated through different polarization angles. At pol. angle Φ = 0°, the PMA showed the same numerical result as the normal incidence. For Φ = 15°, unity absorbance was found at three absorption peaks: 3.0, 5.7 and 7.2 GHz. Nearly the same absorption was found at 3.02, 5.57 and 6.98 GHz for the polarization angle 30°. Furthermore, absorbance was also investigated for Φ = 45° and 99.84%, 98% and 99.69%, and absorbance showed at the resonance frequencies of 3.13, 5.85 and 7.25 GHz, respectively. Finally, for Φ = 60°, three resonance peaks were observed at 3.03, 5.83 and 7.22, with absorption at 99.63%, 98.82% and 98.29%, respectively. It can be concluded that the projected PMA is polarization insensitive up to 60°, which is shown in Figure 5. 

E-field, H-field and surface current density were analyzed to create a better perception of the operational mechanisms of the three absorption peaks. Figure 6 illustrates the magnetic field distribution at different resonance frequencies. The electromagnetic properties of an MA depend on the charge produced [32]. The motion of charged particles relates to the electric field and magnetic field. Maxwell’s curl Equations (3) and (4) are the best tools to use to comprehend the physical characteristics of the proposed PMA.
(3)$$\nabla \times E={-\mu }_{r}\mu_{0}\frac{\partial H}{\partial t}$$
(4)$$\nabla \times H=-\epsilon_{r}\epsilon_{0}\frac{\partial E}{\partial t}$$
where, *E* and *H* are the electric field vector and magnetic field vector, *µ_r_* and *ϵ_r_* are relative permeability and permittivity.

where vector operator,
$$\nabla =\hat{i}\frac{\partial }{\partial x}+\hat{j}\frac{\partial }{\partial y}+\hat{k}\frac{\partial }{\partial z}$$

The surface current, magnetic field and electric field activities of the proposed PMA can be explained with the help of Equations (3) and (4). Figure 7 depicts the current distribution for three resonance frequencies: 3.03 GHz, 5.83 GHz and 7.23 GHz. The red bold arrow sign indicates the surface current and flow direction. This is observed in Figure 7a, the maximum surface current is concentrated on the outer SRR. Hence, the resonance frequency, *f*_1_ = 3.03 GHz of S-band, is a contribution of the outer SRR resonator. Since the metallic resonators have the effects of an inductor, a coupling effect may occur between the two SRRs. With the increase of the frequency, the surface current gradually moved to the inner resonator’s part of the unit cell. Figure 7b shows that the second SRR was responsible for the mid resonance frequency, *f*_2_ = 5.83 GHz. At resonance frequency, *f*_3_ = 7.23 GHz, surface current density was high at the circular split-ring resonator (CSRR) and gradually spread to the swastika shape, which is shown in Figure 7c. According to the Ampere and Biot-savart theory, induced magnetic field strength is a function of current flow (I) and the distance (r) from the conductor [33] as stated in Equations (5) and (6).
(5)$$B=\frac{\mu_{0}I}{2\pi r}\hat{r}$$
(6)$$\mathrm{or}\;B=\frac{\mu_{0}q}{2\pi rt}\hat{r}$$

Figure 6 illustrates the magnetic field distribution at three resonance frequencies of the proposed PMA. It was observed from the pattern of the magnetic field that the intensity of H varied with the intensity of surface current and declined from the higher to the lower values, which are shown in the circles. The induced magnetic fields interacted with one another, the weak magnetic field was induced at the center of the metal rings and the changing magnetic fields generated the curling or twisting tendency of the electric field, as mentioned in Equations (3) and (4). A similar effect was noticed in the case of electric field distribution, which is illustrated in Figure 8. By comparing the two fields, it is vividly observable that the intensity of the E-field was low at the point where the magnetic was also low, and the position of the two fields was perpendicular to each other. It was also observed that the E-field intensity in the split gaps is greater because every split gap in the resonator has created the capacitance effect, which gathers more charge to produce an additional E-field. 

In this section, the fundamental mechanism of EM radiation absorption will be explained.

In the first resonance frequency, 3.03 GHz, the charges produced in the outer SRR moved in a left to right (clockwise) direction. As a result, large numbers of positive and negative charges accumulated at the two opposite terminals of the concerned resonator. Similar effects were observed for the rest of the two resonance frequencies, 5.83 GHz and 7.23 GHz, at the respective part of the resonators. This anti-parallel surface current flow generated the magnetic dipoles which created the magnetic resonance. The incident electromagnetic radiation was absorbed by this magnetic resonance. Some contemporary published articles are compared with the proposed PMA in Table 2.

### 3.1. Design Selection Technique 

The best responsive design was selected for the expected outcome through a trial and error method. Moreover, the resonance frequency shifting factors were marked through the surface current analysis to ensure the anticipated frequency bands [40].

An SRR, with dimensions of 9.6 × 9.6 × 0.035 mm^3^, was placed on the substrate along with a swastika shape, which is shown in Figure 9a. In a numerical simulation, it yielded dual resonance peaks at 2.4 GHz and 5.94 GHz with an absorbance of 93.70% and 92.27%, respectively. Another SRR of 8.2 × 8.2 × 0.035 mm^3^ was inserted inside the first figure that exhibited triple resonance frequencies at 2.05, 3.12 and 5.93 GHz with an absorbance of 24.60%, 99.07% and 98.33%, respectively [shown in Figure 9b]. Figure 9c, the third Figure in the design steps, includes a circular split-ring resonator (CSRR), which yielded four resonance peaks at 2.05, 3.11, 5.91 and 7.75 GHz with an absorbance of 24.05%, 99.03%, 97.48% and 21.79%, consecutively. Finally, the circled swastika shape was repeated in the rest of the three quadrants through a 90° rotation of the phase shift for each quadrant, which is depicted in Figure 9d. In the numerical simulation it exhibited triple resonance peaks at 3.03 GHz, 5.83 GHz and 7.23 GHz with nearly unity (≈100%) absorbance, which met the expected outcome of the proposed project.

### 3.2. Parameter Studies of the Proposed PMA Unit Cell

Next, various parameters were studied to achieve the expected outcome from the proposed design. In this regard, the widths, split gaps and material changing effects on the numerical results were observed.

*i**.* 
*Width Changing Effect*


Firstly, a width changing effect was observed for two split-ring resonators, which remained unchanged regarding the other parameters. Three absorption peaks were observed for the 0.30 mm and 0.40 mm width at 3.18, 6.02 and 7.47 GHz with absorption at 99.84%, 97.17% and 99.60%, respectively. For the 0.50 mm width, the numerical result also showed three resonance peaks at 3.03, 5.83 and 7.23 GHz with 99.84%, 99.03% and 98.26% absorbance, respectively. Similarly, for the 0.60 mm width, it found that it exhibited three resonance peaks at 2.84, 5.6 and 7.29 GHz with 98.48%, 92.09% and 96.12% absorption, respectively. The absorption results for these three widths of two SRRs is depicted in Figure 10a.

*ii**.* 
*Effect of Split Gap*


The split of the resonators had great influence on absorption. The numerical results were observed for the three consecutive split gaps of two SRRs: 0.30 mm, 0.40 mm and 0.50 mm. For the split gap 0.30 mm, three resonance peaks were observed with an absorbance of 99.90%, 99.15% and 97.39% at resonance frequencies 3.01, 5.73 and 7.20 GHz. For the split gap, 0.40 mm exhibits three resonance frequencies, 3.02, 5.82 and 7.25 GHz, with an absorbance of 99.84%, 98.76% and 97.72%. Lastly, for the width 0.50 mm, three absorbance peaks were also observed at 3.03 GHz, 5.83 GHz and 7.23 GHz with absorption at 99.84%, 99.03% and 98.26%, respectively. Figure 10b represents the absorption results for the three split gaps.

*iii**.* 
*Material Analysis*


The proposed PMA was also optimized for different materials. A study was carried out for two Roger’s materials, RT 5870 and RT 6002, and for the epoxy resin binder material FR 4, which is depicted in Figure 10c. The numerical absorption result for RT 5870 exhibited three resonance peaks at 2.54, 3.83 and 7.02 GHz with absorption at 77.23%, 43.44% and 45.63%, respectively, whereas RT 6002 showed very low absorption at the three resonance frequencies. Similarly, material FR 4 yielded three resonance frequencies at 3.03, 5.83 and 7.23 GHz with 99.84%, 99.03% and 98.26% absorbance, respectively. Hence, the unit cell of the proposed PMA is configured with the width at 0.50 mm, a split gap 0.50 mm and a FR 4 dielectric.

### 3.3. Results Validation

*i**.* 
*Equivalent Electrical Circuit Simulation by ADS*


The numerical result of the reflection coefficient (S_11_) for the unit cell of the offered PMA was validated by advanced design system (ADS) software version 2016. The unit can be considered as an LC tank as it is a combination of metal strips of different shapes with some split gaps [41]. The metal strips and the split gaps play the roles of inductors and capacitors, respectively, which are responsible for creating three resonance peaks. These resonance frequencies were determined with the help of Equation (7).
(7)$$n=\frac{0.159}{\surd (LC)}$$
where, *L* = inductance (µF) and *C* = capacitance (pF).

Again, the capacitance can be calculated by using Equation (8)
(8)$$C=\in_{o}\in_{r}A\frac{1}{d}$$
where, *ϵ_o_* and *ϵ_r_* denote the permittivity of free space and relative permittivity of the medium, *A* = area of the metal strip and *d* = split gap distance.

The equivalent capacitance of the circuit (*C_eq_*) can be derived by using Equation (9).
(9)$$C_{eq}=\in_{o}t[\frac{1}{2\pi }\left\{\frac{2w}{d+h}+\frac{\left(g+h\right)}{d+h}\right\}ln\left\{\frac{2(d+g)}{a-l}+\frac{\left(g+h\right)}{a-l}\right\}$$

Furthermore, equivalent inductance of the circuit can be estimated as [42,43].
(10)$$L_{eq}=\frac{{µ}_{o}}{100}\left\{\frac{{2(d+g+h)}^{2}}{(2w+{g+h)}^{2}}+\sqrt{\frac{{\left(2w+g+h\right)}^{2}+l^{2}}{{\left(d+g+h\right)}^{2}}}\right\}t$$Here, the value of permittivity in free space and permeability in free space is $\in_{o}={8.854\times 10}^{-12}$ Fm^−1^ and ${µ}_{o}={4\pi \times 10}^{-7}$ Hm^−1^, respectively, and w, h, t and l are used for conducting patch width, depth of the substrate, patch thickness and length of the conductor, respectively.

Figure 11a depicts the equivalent electrical circuit of the projected PMA unit cell. The upper patch of the unit cell consists of three types of resonators, such as the outer two SRRs, the circular split-ring resonator (CSRR) and the swastika-shaped metal patch. L1, L2, L3, L4, L5, L6, L7 and L8 are the inductors while C1, C2, C3, C4, C5, C6, C7, C8, C9, C10, C11, C12 and C13 are the capacitors. The first two resonance frequencies, 3.03 GHz and 5.83 GHz, were created by L1, L2, C1, C2, L3, L4, C3, C4 and the coupling capacitor C5, which were used for two SRRs. The swastika-shaped resonators and their CSRRs are presented by (L5, C5), (L6, C6), (L7, C9) and (L8, C10), whereas C8, C11 and C12 are adjoin capacitors that contribute a higher resonance frequency: 7.23 GHz. Lastly, the inductors L9 and L10 were used for bottom copper, and the C13 capacitor coupled them with the top layer. The LC circuit was authenticated through simulation with a path wave advanced design system (ADS), and the values of Ls and Cs were tuned to gain the expected resonance frequencies and satisfactory magnitude. The comparison of the CST and ADS results of the reflection coefficient (S_11_) for the unit cell shown in Figure 11b, show analogous patterns with insignificant discrepancy. This little inhomogeneity of results happened because the equivalent circuit was constructed with the lumped elements of the unit cell, which were manually tuned to gain an approximate result.

*ii**.* 
*Array Simulation*


The numerical results of absorption for 1 × 2 and 2 × 2 exhibited almost identical results with the CST results of the unit cell, which is depicted in Figure 12. It can be observed from the figure that the array results are slightly moving toward the positive end of the x-direction. A slight influence of the coupling effect can be noticed in the array results.

*iii**.* 
*Prototype Fabrication and Measurement*


Figure 13 shows the prototype measurement setup and measurement result in the microwave laboratory. Glass-reinforced epoxy-laminated dielectric substrate FR-4 with two side copper layers was used to fabricate the unit cell, 1 × 2 and 2 × 2 array of unit cell of proposed PMA for experimental measurement. 

Figure 13a depicts the scaling of the unit cell and Figure 13b shows the PNA with two waveguide ports. For the three different resonance frequencies, various frequency range waveguide ports were used, such as 112WCAS, 137 WCAS and 187WCAS. The fabricated prototype of the different array was placed in the middle of the waveguide ports, which is shown in Figure 13c. The scenario of using a reflection coefficient (S_11_) for measurement and CST simulation is shown in Figure 14a, whereas Figure 14b shows the simulated and measured absorption. We noticed that the measured result slightly deteriorated from the simulated result with some ripple. There are many factors that may have affected the result, such as metallic copper deterioration, coupling impact of array or the air column between the two waveguide ports. In addition, the absorption results may also have been affected by the following reasons: There might be a calibration error associated with the Agilent N5227 vector network analyzer (by Agilent N4694-60001 Ecal), a mutual resonance effect between the transmitting and receiving ends of the waveguides or it could have been a fabrication error in the designed metamaterial. In addition, results also can be affected by the permittivity of the substrate because the substrate was an important factor in the variation between simulated and measured results. The resonance frequency depends on the substrate material’s permittivity. If the permittivity increases, the resonant peaks are shifted toward lower frequencies. When the dielectric constant is increased, the capacitance values between the ground and each of the radiating elements are also raised. Therefore, each capacitance associated with the radiating elements is in series with the others, and the system’s equivalent capacitance is reduced. This decrease in the equivalent capacitance is another possible reason for the variations between the results.

## 4. Conclusions

A circular split-ring resonator (CSRR) surrounded by a swastika-shaped triple-band metamaterial absorber has been presented for satellite-dependent telecommunication systems in the S- and C-band. There are three types of resonators that have been used in this design. Each part of the resonator shows a balanced surface current distribution that has a great impact on creating triple resonance peaks. For the authentication of performance of the proposed MA, the simulated results are compared with the prototype measurement results. In spite of the positive evidence regarding this study, some limitations also persist in this design. To minimize the concern about errors, further studies could be conducted based on the inverse relationship of widths, thickness and shifting of resonance frequencies for the purpose of being more versatile. The unity absorption rate signifies the great potentiality of the proposed MA that could be deployed for telecommunication via satellite feeds.

## Figures and Tables

**Figure 1 materials-16-04776-f001:**
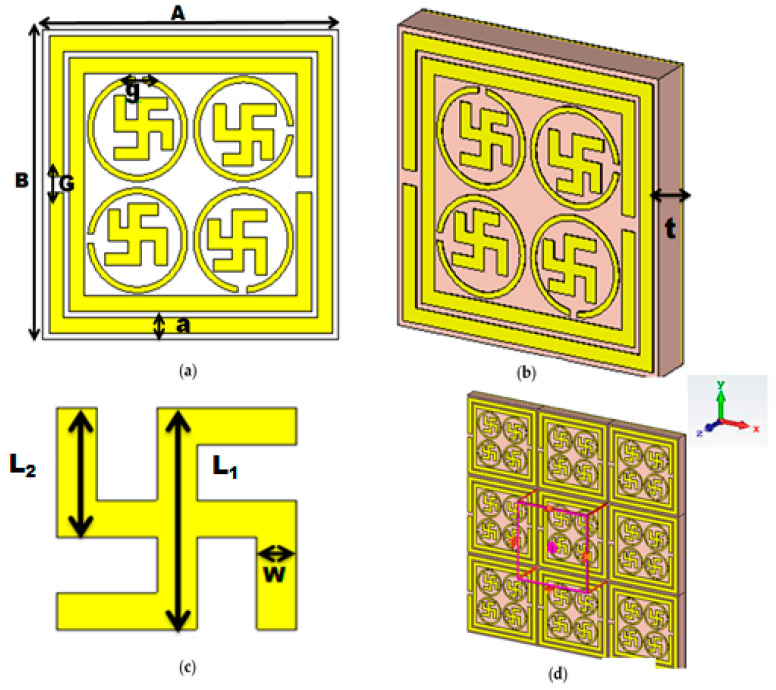
(**a**) Top view of unit cell (**b**) oblique view (**c**) swastika shape (**d**) simulation view.

**Figure 2 materials-16-04776-f002:**
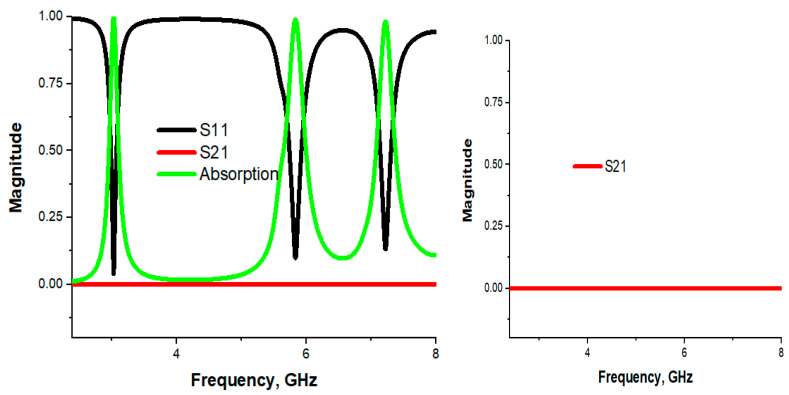
Graphical view of scattering parameters (inst S_21_).

**Figure 3 materials-16-04776-f003:**
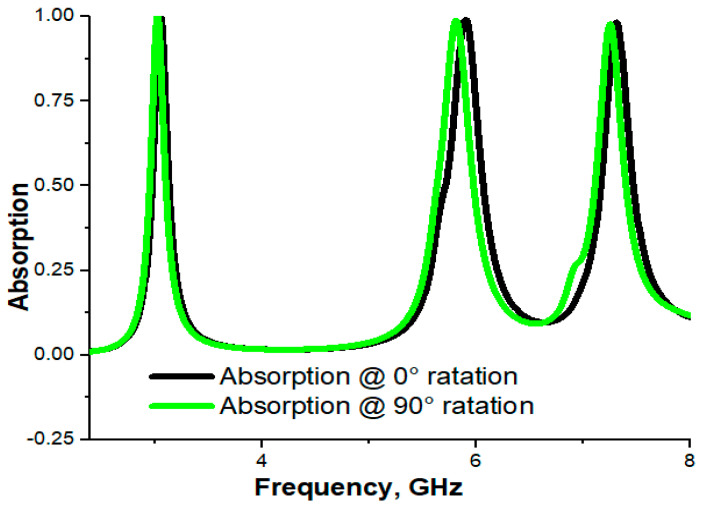
Absorption at different angles of rotation.

**Figure 4 materials-16-04776-f004:**
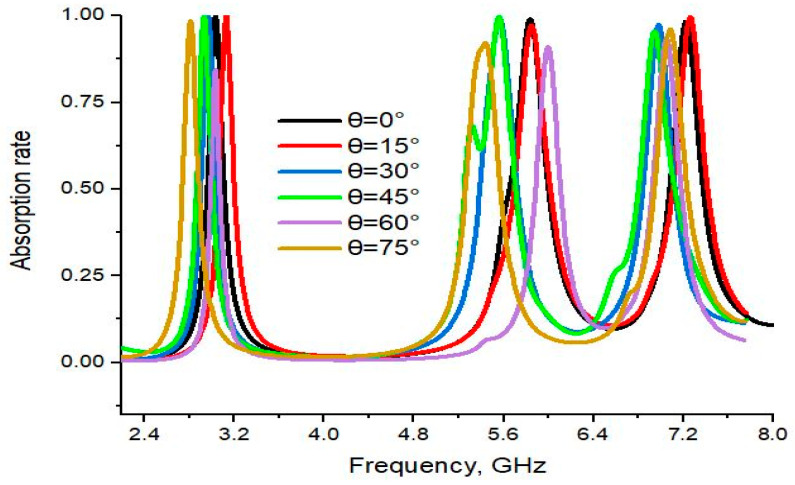
CST results of absorption at various oblique incident angles.

**Figure 5 materials-16-04776-f005:**
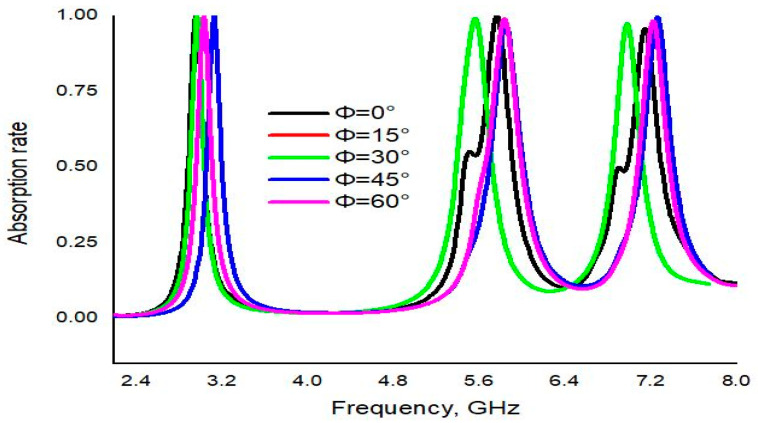
CST results of absorption at different polarization angles.

**Figure 6 materials-16-04776-f006:**
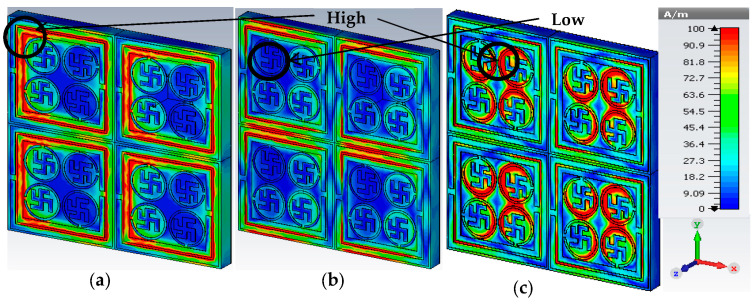
Magnetic field distribution at (**a**) 3.03 GHz, (**b**) 5.83 GHz and (**c**) 7.23 GHz.

**Figure 7 materials-16-04776-f007:**
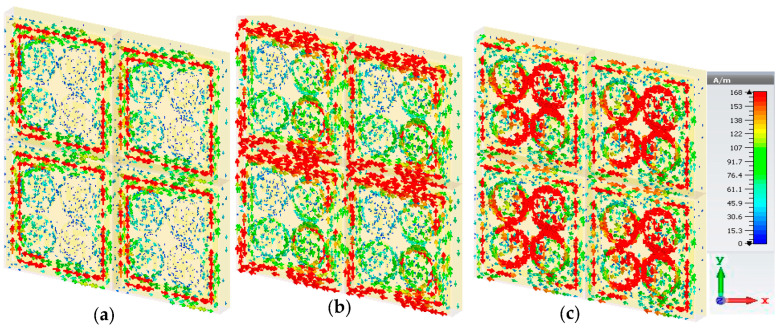
Surface current distribution at (**a**) 3.03GHz, (**b**) 5.83 GHz and (**c**) 7.23 GHz.

**Figure 8 materials-16-04776-f008:**
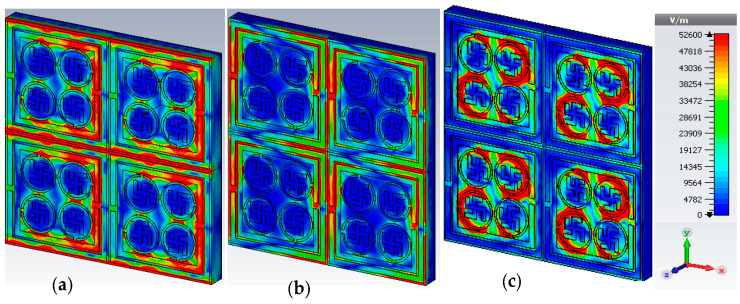
Electric field distribution at (**a**) 3.03 GHz, (**b**) 5.83 GHz and (**c**) 7.23 GHz.

**Figure 9 materials-16-04776-f009:**
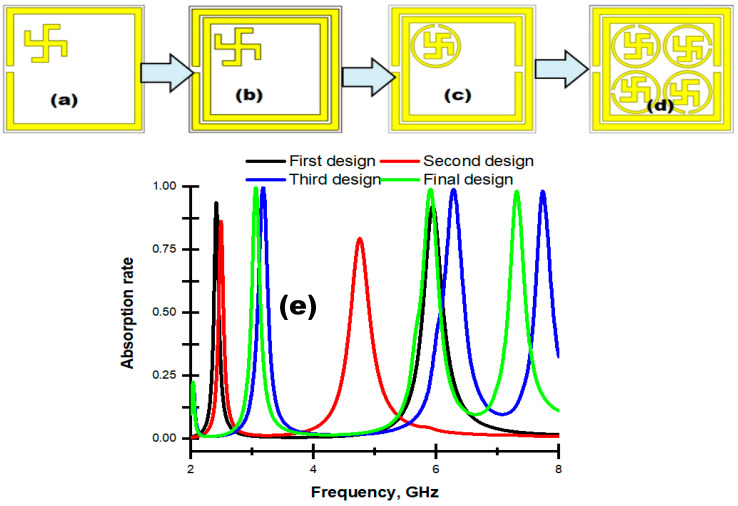
Design selection steps. (**a**) First design, (**b**) second design, (**c**) third design, (**d**) final design and (**e**) absorption for different designs.

**Figure 10 materials-16-04776-f010:**
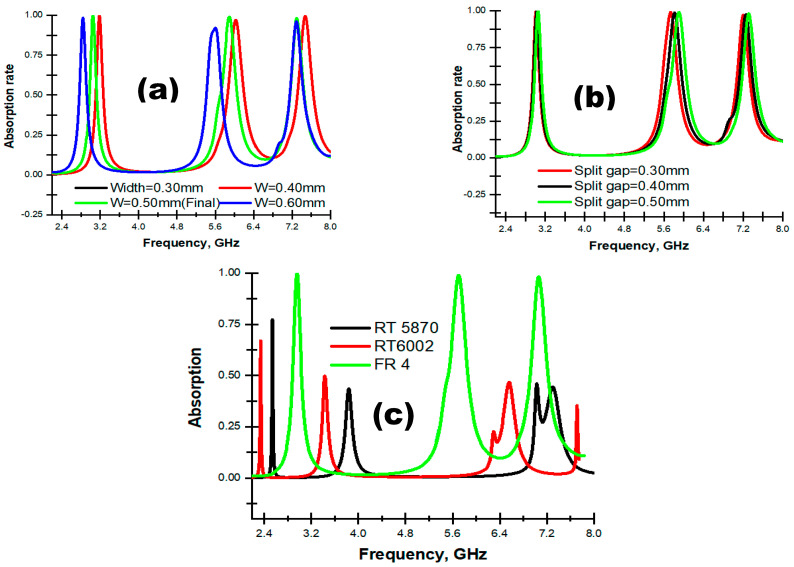
Absorption rate at (**a**) changing widths of SRRs, (**b**) changing split gaps and (**c**) different materials.

**Figure 11 materials-16-04776-f011:**
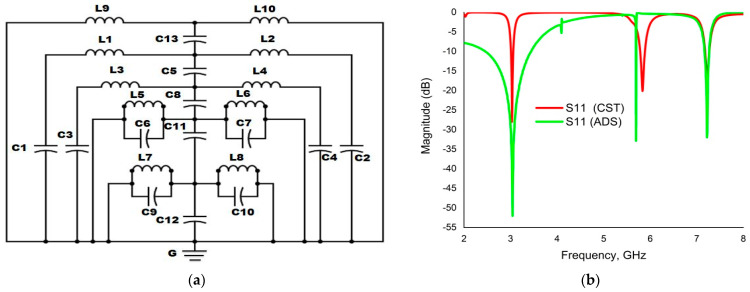
(**a**) LC equivalent electrical circuit (**b**) Comparison of CST and ADS results of reflection coefficient (S_11_).

**Figure 12 materials-16-04776-f012:**
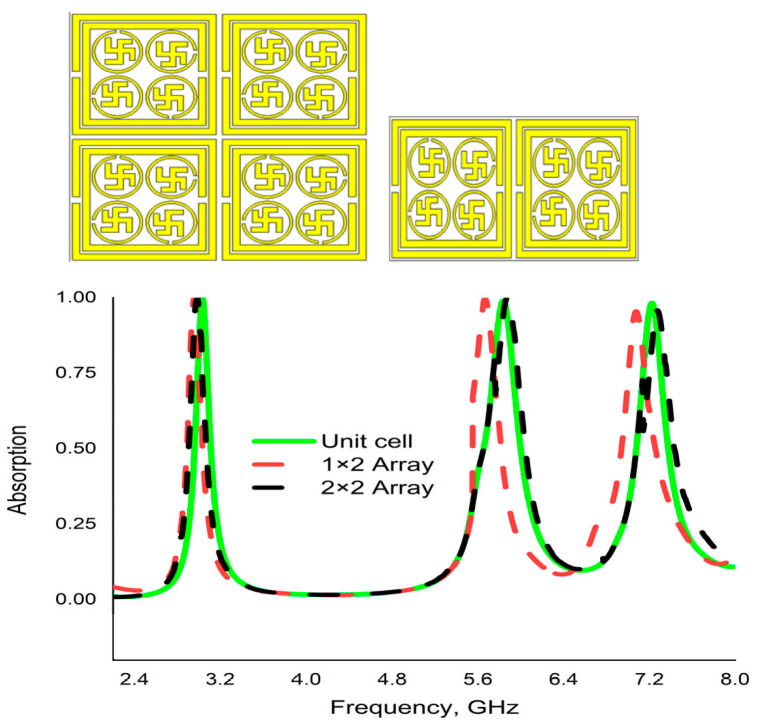
Comparison of numerical absorption results among unit cells, 1 × 2 and 2 × 2 array.

**Figure 13 materials-16-04776-f013:**
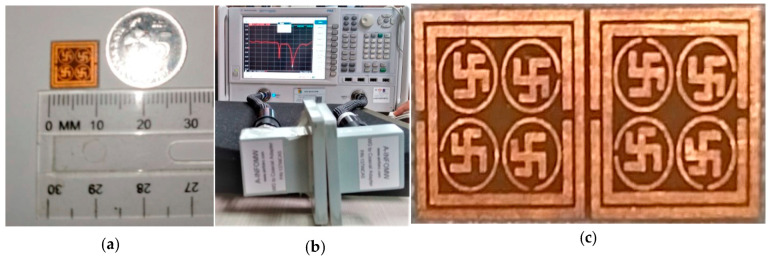
Measurement setup (**a**) unit cell (**b**) microwave network analyzer (PNA) Agilent N5227 (**c**) 2 × 2 array.

**Figure 14 materials-16-04776-f014:**
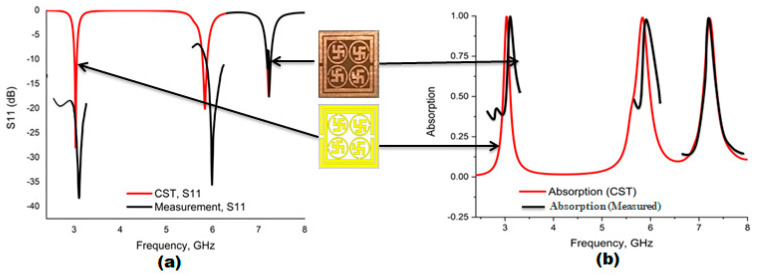
(**a**) Simulated and measured reflection coefficient (S_11_). (**b**) simulated and measured absorption.

**Table 1 materials-16-04776-t001:** Optimized geometric value of the PMA unit cell (all units are in mm).

A	B	t	a	G	t_1_	L_1_	L_2_	g	w
10	10	0.035	0.50	0.50	1.60	2.0	0.8	0.30	0.40

**Table 2 materials-16-04776-t002:** Comparison of proposed PMA with some published articles.

Reference No.	Design Shape	Unit Cell Size (mm^2^)	Covering Band	Frequency at Max. Absorption GHz	Max. Absorption (%)	Published Year
[34]	Dipole-based square ring	11 × 11	S- and X-	10.90	90	2015
[35]	Spike-shaped	8 × 8	X- and Ku-	17	90.90	2020
[36]	Wrenched square	10.4 0 × 10.40	S-, X-, Ku-	11.15	97.69	2021
[37]	Jerusalem cross-shaped	11.50 × 11.50	C- and X-	11.6	97	2021
[38]	Double E-shaped	10 × 10	C- and X-	10.32	99.90	2022
[39]	Modified square-shaped	35 × 32	L-, S-, C-	5.50	99	2023
Proposed PMA	CSRR bounded swastika-shaped	10 × 10	S- and C-	3.03	99.84	--

## Data Availability

All the data are available within the manuscript.

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
