# Peer review of "Compact and Polarization Insensitive Satellite Band Perfect Metamaterial Absorber for Effective Electromagnetic Communication System"

_materials, 2023, doi:10.3390/ma16134776_

Round 1

Reviewer 1 Report

In this manuscript, “Compact and Polarization Insensitive Satellite Band Perfect Metamaterial Absorber for Effective Electromagnetic Communication System,” the authors present a commercially viable MDM configured triple band metamaterial absorber. Overall, this manuscript has a strong potential for another review round after applying the issues and addressing the shortcomings listed below:

1-The authors should polish/revise some grammatical mistakes and typos along the manuscript. I invite the authors to read their manuscript carefully and make the required changes where necessary.

2-Please increase the size of the text provided in the figures (where necessary).

3-Please increase the thickness of the lines within the figures (where necessary).

4-In the Introduction section, while discussing recent developments in the field of absorbers, the following works should also be considered and cited to give a more general view to the possible readers of the work: [(i) Toroidal Metaphotonics and Metadevices, Laser and Photonics Reviews 14, 1900326 (2020), (ii) Conceptual-based design of an ultrabroadband microwave metamaterial absorber, PNAS 118, e2110490118 (2021)].

5-Figure 5 (all panels) seems a bit confusing. Please make the arrow more explicit.

6-In Figure 13, for the experimental data, please add the missing data points (to be able to make a fair comparison between simulation and experimental results).

N/A.

Author Response

As attached.

Reviewer 2 Report

The manuscript presents a high frequency electromagnetic meta-surface work with majority on numerical simulation. Some major comments need to be addressed before the reviewer recommends acceptance.

1.       First of all, the design concept is better to explain. How the specific geometry come up?

2.       For the swastika unit cells, the four cells are axial (rotational) symmetric. Although, for a large-size and isotropic wave source, it should not matter if the swastika unit cells are axial (rotational) symmetric or flipping symmetric, the performance would be different when the wave source is anisotropic or smaller than a square unit cell. Is there any specific reason there? Please explain.

3.       For the numerical simulation, the reviewer has some questions. First at all, the reviewer suggests the authors provide band structure of the unit cell.

4.       For the transmission spectra, it is not common that the transmission is zero over the wide frequency range. Please verify or explain.

5.       As the authors claimed, the bandgap frequency provides strong absorption which following the conventional explanation. Hence, in usual, the absorption is realized by the anti-resonance mode of the unit cells (swastika or rings). However, there was no significant electric field strength on the swastika cells I the numerical results. Please provide some explanation.

6.       For the experiments, the transmission spectra, the discontinuous frequency sweeping is not acceptable for publication. Please provide the complete spectra. 

Author Response

As attached.

Reviewer 3 Report

Comments on the manuscript

This manuscript introduces a commercially viable triple-band metamaterial absorber for effective electromagnetic communication systems. The proposed absorber was designed with a unique, small structure yet has higher absorption than current studies. It is ideal for telecommunication systems that rely on full-time raw satellite and radar feeds. Also, this work includes detailed parameter simulations and equivalent electrical circuit studies.

The manuscript presents intriguing scientific research on microwave metasurface absorbers, contributing to their advancement. The technical aspects of the study are well-executed; however, the explanation of the physics governing angular instability for multi-band absorbers could be further elaborated, enhancing the overall quality of the paper. Despite this, the work is suitable for publication in Materials as long as minor issues are addressed.

1.      Quoting “Due to it’s novel and excellent characteristics, it is deploying in microwave region [7], terahertz region [×], infrared [9] and visible wavelength region [10].” This statement is correct. However, there are two imperfections in this sentence. First, a typo, [×]. Second, some recent advances in the infrared range are missing. For example ["Bound states in the continuum in anisotropic plasmonic metasurfaces." Nano Letters 20.9 (2020): 6351-6356; "Dual-band perfect absorber for a mid-infrared photodetector based on a dielectric metal metasurface." Photonics Research 9.1 (2021): 27-33].

2.      The author stated that this meta device has a deep subwavelength lattice spacing, with an electrical dimension of the unit cell measuring 0.101λ×0.101λ, making it very compact in size. Figure 3 shows the angular instability for various oblique incidences, and the underlying physics that causes this phenomenon is not clear. Please comment.

3.      About Figure 13(b), the authors note that while the simulated and measured absorption results are similar, there is some deterioration in the measured result with ripple. They suggest that multiple factors, such as metallic copper deterioration, coupling impact of the array, and the air column between two waveguide ports, may have affected the results. However, it is unclear whether the authors considered the finite size effects and the impact of incidence angle span in their analysis.

readable

Author Response

As attached.

Reviewer 4 Report

The manuscript by Afsar et al. presents a metal-dielectric-metal configured triple-band metamaterial absorber. The absorber is an assembly of four symmetric circled compact swastika-shaped metal structure that are bounded by two split ring resonators (SRRs). The structural parameters of the unit cell are determined by trial-and-error method. Three resonance peaks are observed at frequencies 3.03, 5.83 and 7.23 GHz with absorbance of 99.84%, 99.03% and 98.26%, respectively. The proposed PMA has unique design and small dimensions with higher absorption compared to the contemporary studies. This special type of polarization-insensitive S- and C-band PMA is designed for telecommunication systems via full-time raw satellite and radar feeds. Overall, I found this study to be very comprehensive, the claims are well supported, and the conclusions appear to be sound. Overall, I suggest minor revision, and my main suggestion for improvement as well as suggestions for the authors are listed below:

1.     This work is rather numerical-heavy, although the authors have prepared a prototype containing 1x2 and 2x2 array, this still only represent a small portion of the proposed structure. I wonder if the authors could experimentally validate their results using larger sample sizes, e.g., ones that contain 3x3 ~ 5x5 unit cells.

2.     In the beginning of the abstract, the authors promised a commercially viable triple band metamaterial absorber, which set up a high expectation for me. Although this is not strictly necessary, I would still encourage the authors to discuss the current challenges of fabricating commercially viable metamaterial absorbers, and how this work could be superior than the existing examples.

The overall quality of the language is acceptable, though the English can be improved, mainly in the grammar and style. 

Author Response

As attached.

Round 2

Reviewer 1 Report

In its current form, the revised manuscript is suitable for publication.

Reviewer 2 Report

The authors addressed the comments well.